# Symbolist Androgyny: On the Origins of a Proto-Queer Vision

Damien F. Delille

Laboratory LARHRA, Department of Art History & Archeology, University of Lumière Lyon 2,
Lyon 69007, France; damien.delille@univ-lyon2.fr

**Abstract:** This article focuses on artistic and aesthetic practices within the idealist and symbolist movements of the late 19th century in France. It investigates how artists and art critics embraced androgynous imaginaries derived from Greco-Roman antiquity and the Platonic myth, transforming them into tools for social and sexual emancipation and giving rise to a proto-queer vision. An analysis of the art of Alexandre Séon, Odilon Redon, Jeanne Jacquemin, and Léonard Sarluis, in conjunction with the symbolist theories of Joséphin Péladan, Gabriel-Albert Aurier, and Émile Verhaeren, reveals an idealistic pursuit grounded in the union of the masculine and the feminine through the act of creation. Through the examination of artworks, contemporary critical discourse, and the personal correspondence of these art figures, this study posits that the androgyne serves as a heuristic model for a queer art history. The ideal androgyne, as theorized in Freud's psychoanalytic writings, can function as a methodological paradigm in art studies as a tool for visualizing and conceptualizing homosexuality in art.

**Keywords:** androgyny; queer vision; mysticism; homosexuality in art; psychoanalysis





## 1. Introduction

Reinterpreting late 19th-century art history through a queer lens presents several challenges. Beyond the potential anachronism of the term, queer theories are deeply embedded in the political and cultural struggles of the late 20th century that challenged heteronormativity and complicated interpretations of same-sex desires, incorporating into them issues of race and class. The queer art and culture of this period used more fluid representations of gender identity to challenge stereotypes associated with stigmatized minorities. Nonetheless, a century apart, it is possible to draw parallels between these two fin-de-siècle periods, as both were marked by anxiety about the end of times, which was linked to an "end of sex" (Winn 1997). How does the symbolist and idealist art of the late 19th century contribute to contesting gender norms? What role does the private life of artists play in shaping an embryonic queer culture? This study aims to investigate these questions by extending research on androgynous imaginaries in symbolist art, following the pioneering work of Patricia Mathews (Mathews 1999) and building on other developments, which have already been presented extensively in my previous study (Delille 2021). Here, the androgynous model will be analyzed as a proto-queer ideal that seeks to transcend normative masculine and feminine identities challenged by the emergence of feminist and homosexual critiques.

The late 19th century was characterized by paradoxical currents of thought and creation, blending avant-garde ideals with a more conservative vision marked by various forms of spirituality (Marlais 1992). Collaborating with writers and art critics of the time, symbolist and idealist artists envisioned an idealized relationship with the model, taking the form of an androgynous union between the masculine and feminine. This attempt to surpass gender norms rooted in the biological identification of sex echoes social and visual constructs inherent in later queer theories (Lord and Meyer 2013; Davisson 2014), making these proto-queer visions resonate with modern conceptions of homosexuality. As the first theories of psychopathology applied to sexuality emerged, leading to the coining

of the term "homosexual" in German in 1868 (Kennedy 1997), hermaphroditism served as a physiological model to explain sexual deviation (Domurat 1998; Cryle and Forth 2008). The identification of the homosexual, a character invented in the 19th century, according to Michel Foucault (Foucault 1976; Merrick and Sibalis 2001), was scientifically conceived as the psychic union of the two sexes—the theory of hermaphroditism of the mind—and the inverted appearance of the two sexes: the effeminate man and the masculine woman.

These processes of identifying and visualizing gender norms were contested by symbolist artists themselves, who saw the androgynous figure as a provocative means to disrupt visual perception. This study aims to offer a nuanced analysis of the works and statements of these artists to produce a more comprehensive understanding that counters their often essentialist critical reception. Indeed, the symbolist theory of artistic creation relied on a disrupted vision of the relationship between mimetic representation and artistic inspiration. Following Whitney Davis's analyses on the dialogue between classical aesthetics and homoeroticism (Davis 2010), this study seeks to show how the androgyne, whether depicted or fantasized, is a heuristic model. Androgynous imaginaries evoke older forms of non-normative sexuality, challenging the bourgeois, materialistic conception of the family. While Oscar Wilde's 1895 trial made visible the assertion of homosexual pride associated with art, this period also harbored other underground currents with ramifications for an emerging queer history of art.

### 2. *Curieuse!* Androgynous Obsessions in Idealist and Symbolist Circles

The original myth of the androgyne from Plato was appropriated by symbolists, who were driven by romantic nostalgia for the search for the soulmate. The art critic and organizer of the Rosicrucian Salons de la Rose † Cross, Joséphin Péladan, indicated this in his 1890 novel *Androgyne*, a series of "reconstructions of Greek ephebic impressions through Catholic mysticism" (Péladan 1890a, p. 20). The return to the ancient myth involved an androgynous model, a symbol of the fusion of sexes that brought together Greco-Roman antiquity and mystical Christianity. In Péladan's novel, a young boy becomes a martyr to the ideal, after being subjected to the prevalent materialistic environment that the author calls "ochlocratic" (from the Greek -*okhlos*, the crowd, and -*kratos*, power: the political power of the crowd). Péladan's epic aligned with a wave of conservative, Catholic, and monarchist reaction following the political events of the Boulanger Affair in France in the early 1880s (Marlais 1992, pp. 55–56; Beaufils 1993). In contrast to the realist art of Édouard Manet and Gustave Courbet, Péladan believed that the artist must withdraw from the world and attain an androgynous ideal through spirituality.

In his review of the sculpture exhibitions at the Salon des Beaux-Arts in 1883, Péladan identified the ephebe of Polykleitos as the ultimate model: "'He condensed his art into one work,' says Pliny, which amounts to this: 'He condensed man and woman into a man-woman'" (Péladan 1883, p. 433). This condensing of male and female originated in hermaphroditic statuettes in the Greek islands and continued through painting and stage representations throughout history. From Antoine Watteau's *L'Indifférent* (1717) to Anne-Louis Girodet's *Le Sommeil d'Endymion* (1791), these ambiguous figures mentioned in his article (Péladan 1883, p. 440) reminded Péladan of cross-dressing roles in the theater, such as the feminine character of Mignon, based on Goethe's novel *Wilhelm Meister's Apprenticeship* (1795–96), and characters in novels, such as Théophile Gautier's *Mademoiselle de Maupin* (1835) and the Goncourt brothers' *Renée Mauperin* (1864).

Beyond transcending gender norms, the androgyne represented a suspension of eroticism achieved through the spiritual purification of art. Sexuality signified the death of the ephebic ideal of androgynous beauty that Péladan portrayed in his novels. His 1890 novel *Curieuse!* recounts the platonic love of the artist Nebo for the heroine Paule de Rizan. The frontispiece originally commissioned for the novel, drawn by Alexandre Séon, an artist from the Forez region in France, is an enigmatic portrait (Figure 1). This figure is inspired by the haunting gaze of Nebo, a fictional metamorphosis of Péladan, during his encounter with Paule: "The head seemed as small as those of Michelangelo: the features



were accentuated with unforeseen flatness: one could see the modeled aspects with Vinci's multiple planes" (Péladan 1890b, p. 6). This description is reflected in Séon's use of fine hatching to veil the "gaze in a disturbing chiaroscuro" (p. 6). In the novel, Nebo asks Paule to pose with hands clasped under her chin and wearing a "black velvet dress with a modest neckline," facing forward, as if engaging in introspective conversation. While drawing inspiration from the novel, Séon envisioned a unique composite figure: the figure did not have Paule's hair, "helmeted with gold, simply twisted and pulled back on the top of the head (…)," but Nebo's "chestnut hair that must have been blond" (Péladan 1890b, p. 7).

In this portrait, Renaissance inspiration merges with the romantic union of male and female, leading to an aesthetic complementarity between inspiration and artistic creation. The fantasy of Nebo and Paule's union, of the union of the artist and his model, is reflected by this exclamation from the novel: "'Androgyne!—and you yourself!' They judged each other with these words". The title *Curieuse!* evokes Platonic inspiration, free from romantic love or carnal desire between man and woman. The artist and the muse must maintain an ideal connection achieved through celibacy, and thus far from the bourgeois drama of unrequited love depicted in 19th-century literature and art (Prince 2002). "Curious!" also serves a declamatory purpose, inciting the artist and muse to explore their changing states of identity.

However, Séon's illustration was ultimately not chosen, the publisher preferring a more provocative depiction by Félicien Rops. Other works by Séon were accepted during this period, however, such as the frontispiece for Péladan's *La Victoire du mari* published in 1888. Inspired by Wagnerian opera, the novel recounts the saga of the winged knight Adar, sword in hand, traversing various physical and mental states in fin-de-siècle Paris (Péladan 1888). The writer's prose shows the influence of esoteric and mystical treatises, offering a more complex vision of the androgyne as an artistic creation of the future. Séon's drawing (Figure 2) emphasizes enigma and mystery, with an Egyptian Sphinx in the background and the indeterminate features of the hero facing his destiny. Commissions such as these engaged artists in a dialogue with mystical writings and symbolist prose. Using a coded language derived from the iconography of myth and religion, they invented figures that highlight transitional states between different times and genres. The perceptual disturbance arises from the difficulty of recognizing the gender of the figures depicted.

From the late 19th century, the term "symbolism" took on an unexpected meaning (Facos 2009). Subjectivity was at the heart of artistic debates of the time, particularly the distortion caused by the style of the impressionists. According to Jean Moréas, symbolism in literature expresses the ambiguity and obscurity of the forms and figures created by the artist's unique vision. In his famous manifesto of 1886, Moréas stated that "in paintings of nature [or] actions of humans, all concrete phenomena cannot manifest themselves; these are sensory appearances intended to represent their esoteric affinities with primordial Ideas" (Moréas 1889). Departing from the realism of Platonic mimetic representation, symbolism adopted Neoplatonic idealism, establishing the symbol as a bridge between the abstract concept and its sensory transmission. Moving beyond Platonic dualism—the strict opposition between man and woman, being and non-being—symbolism reflected the disappearance of certainties, of essences in favor of ephemeral, ever-evolving identities.

This ideal continued in the writings of the young critic Gabriel-Albert Aurier, who in 1891 formulated a vision of symbolism well-known in the history of art criticism. An unexpected model, that of alchemy, which has been extensively analyzed by Patricia Mathews and Juliet Simpson (Mathews 1984; Simpson 1999), merits special attention here. The critic used alchemy to "make material" the new relationship between the artist and his work. Although Aurier did not directly mention the androgynous model, it is possible to trace its symbolist significance through the critic's unitary vision of perceiving and conceiving the world. Distinct from the Platonic differentiation of idea and sensation, an idea is transmitted for Aurier through the evocation of symbols and not through the representation of identified forms. Close to spiritualist ideas positing the existence of an immaterial world, Aurier envisioned an unfathomable interface with reality. In the

opening of his article "Les peintres symbolistes", he quoted Plotinus's *Enneads*: "We attach ourselves to the outside of things, ignorant that what moves us is hidden inside them" (Aurier 1995, p. 95). Unlike conventional allegory, the symbol derives its significance from the viewer, who guides, directs, and completes its meaning through the subjectivity of the artist.

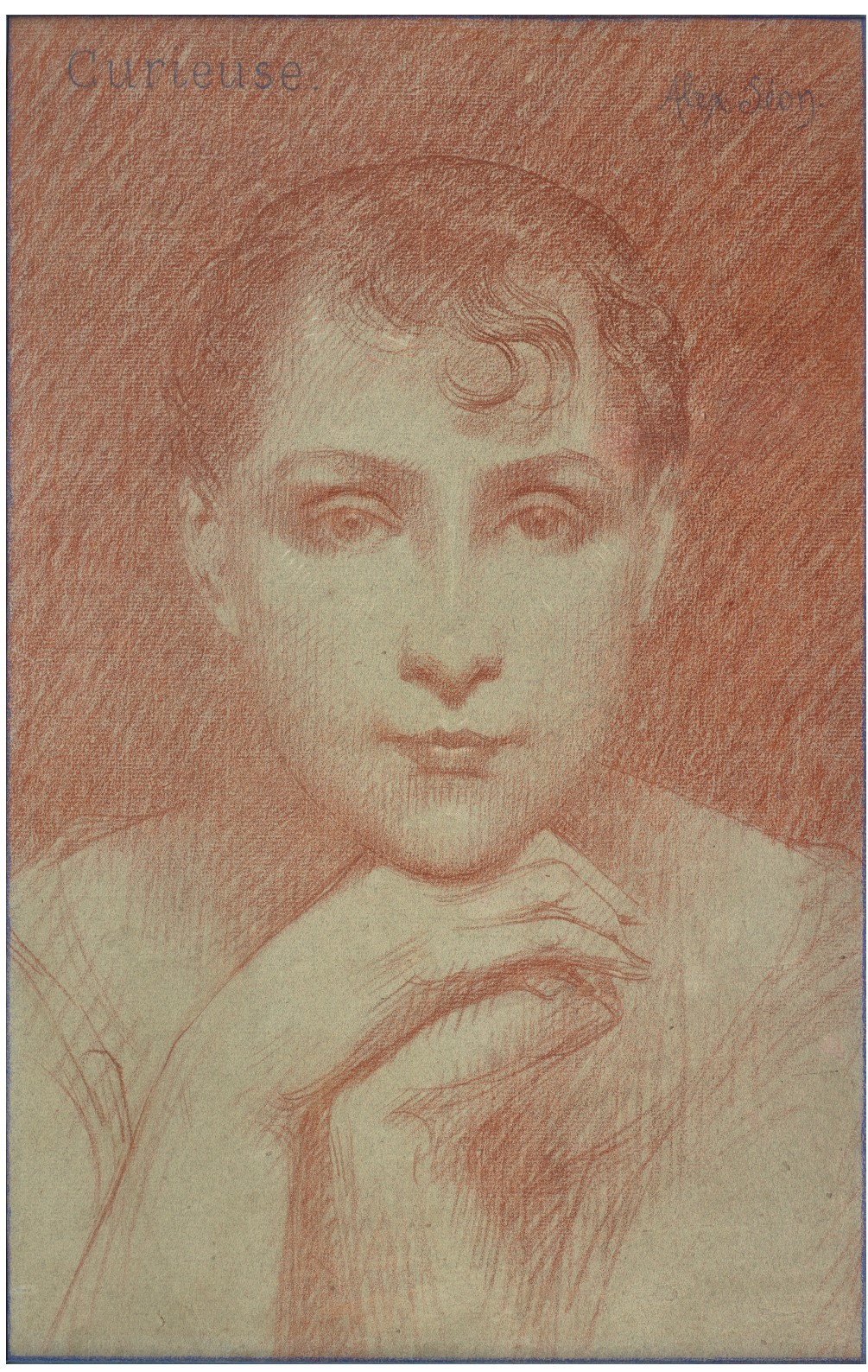

**Figure 1.** Alexandre Séon. *Curieuse*. 1892. Red chalk on paper. 39 × 26 cm, private collection, Lucile Audouy.

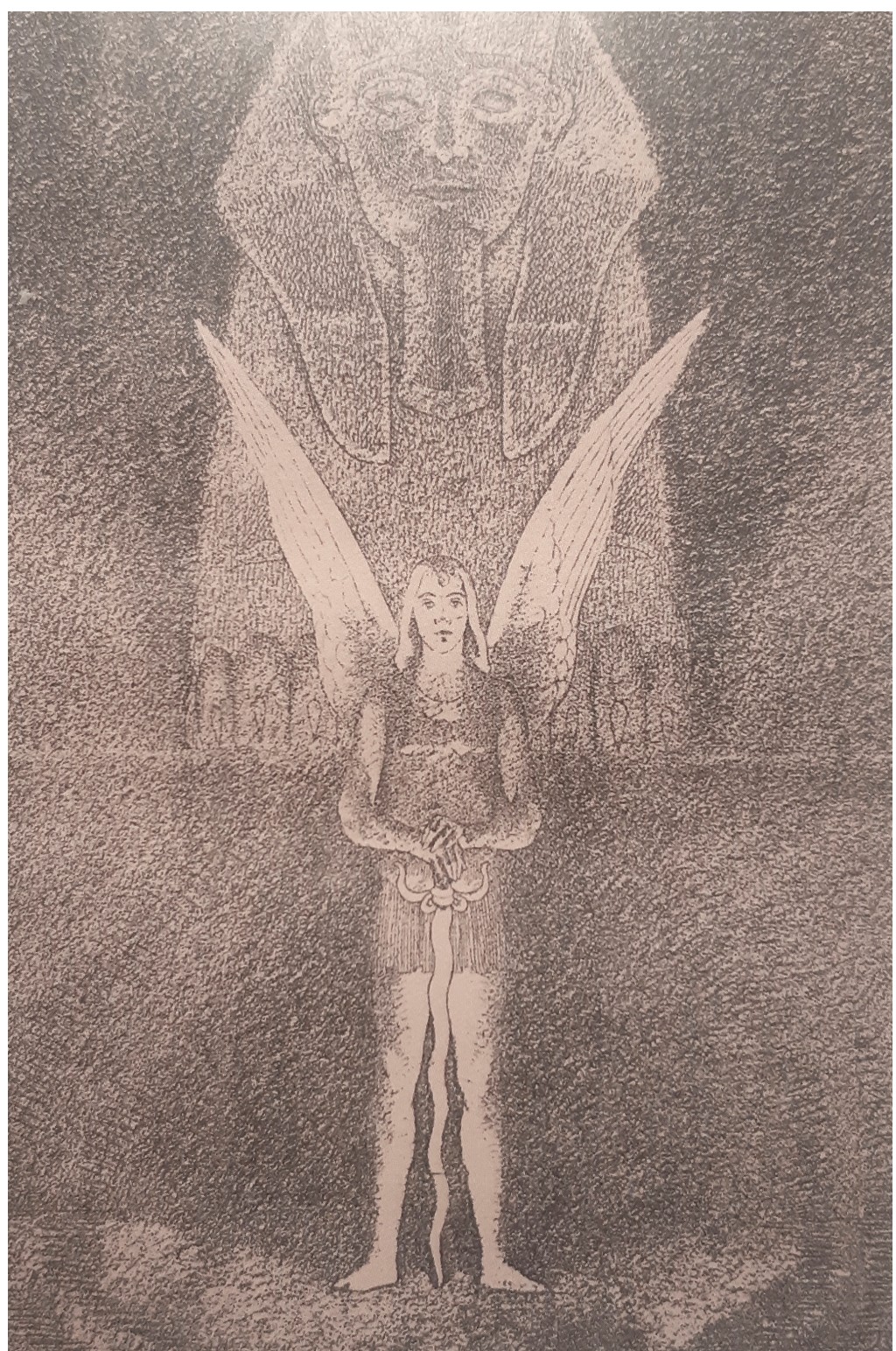

**Figure 2.** Alexandre Séon, *La Victoire du Mari*. 1889. Reproduction. Private collection.

Alchemy serves as a means to materialize the aesthetic relationship, but also to model the symbolist act of creation (Biedermann 1986). For Aurier, the stages of the transmutation of metal serve as a poetic model for the artistic creation of colors and lines. The artist creates new forms inspired by abstract ideas, just as the alchemist forms a new body, the homunculus, a miniature man formed in a flask. Like Prometheus, the artist begets an autonomous form, a symbol born of the creation between nature and the artist's subjectivity:

"A work of art is a new being that not only has a soul, but a double soul (the soul of the artist and the soul of nature, [the] father and mother)" (Aurier 1995, p. 24). Two poles are structured around an essentialist metaphor of the creative act: masculine creation coupled with the feminine nature that inspires it, generating a double or androgynous being, who is brought to life by the spiritual transmutation of the material.

The metaphor of the alchemical androgyne can be likened to the symbolic link between art and the person who contemplates it. Love guides our gaze, according to Aurier, because "the only way to understand a work of art is to become its lover" (p. 95). The symbolist must be detached from a sensual or fantasized relationship with the artwork, which, as with women, would corrupt the abstract, quasi-mechanical aesthetic connection between the symbolist artist and critic. The aesthetic relationship is an act of spiritual union between the masculine and the feminine: "This influx—this sympathetic radiation felt at the sight of a masterpiece, which is called the sense of beauty, aesthetic emotion, and this feeling and this emotion, thus explained by the communion of two souls, one inferior and passive, the human soul, the other superior and active, the soul of the work—will undoubtedly appear, to those who sincerely want to delve into it, very similar to what is called: Love. Even more truly Love than human Love, always tainted with some muddy sexuality. Understanding a work of art is ultimately to love it with love, to penetrate it, I will say, at the risk of easy mockery, with immaterial kisses" (104–5). The ethereal and immaterial worlds, containing primordial forms such as the sphere, correspond to this symbolist ideal. These symbols recall Aristophanes' narrative in his "Eros androgyne" and pave the way for absolute forms of art. These abstract signs are what symbolists must represent, encouraged by Aurier in his rallying cry: "Let us become the mystics of art" (p. 25).

### 3. From Living Model to Androgynous Reinvention

The conception of androgyny within symbolism as developed by Aurier is manifested in painting through the "potential images" (Gamboni 2002) crafted by artists. The gender ambiguity identified in symbolist imagery originates from the intricate connections the artist maintains with the model. The development of painted portraits, whether commissioned from a sitter, or freely inspired, underwent an evolution with the rise of photography and *carte-de-visite* portraiture from the 1850s. The artist no longer needs to offer a resembling portrait that can be used for social identification with the person painted, but can explore the symbolic, even abstract potential of colors and lines induced by the practice of painting. The ambivalence of symbolist aesthetics disrupted the understanding of gender norms. This is evident in the work of Odilon Redon, who initially adhered to a fantastical aesthetic with his early series of *Noirs* illustrating the works of Edgar Allan Poe and William Shakespeare (Rapetti and Redon 2011). From the early 1890s, a chromatic shift toward pastel colors occurred in his work.

Exhibited at the Brussels Salon des XX in 1890, Redon's *In Heaven* (Figure 3) served as a prelude to various artworks employing the androgynous aesthetic. The head of the feminine figure, with closed eyes and short hair, is haloed with blue and yellow brushstrokes. The face and bust, revealing the tops of the breasts, are treated with subtle effects of iridescent yellow, brown, and pink. Set against a depthless violet background, the bust is cut off at the base by a yellow-green horizontal band, creating the impression of its floating in an indeterminate space. Following its success at the salon, Redon produced a lithograph version in fifty copies (Figure 4). In it, the figure was altered to appear more masculine, with an angular jawline, a prominent neck and muscular shoulders, and dark hair flowing down the back. The halo is omitted, and the horizontal band includes vanishing lines, creating the illusion of a solid surface supporting forms of fauna and flora characteristic of the artist's compositions. A third oil version, entitled *Closed Eyes*, was produced the same year (Figure 5); it features a more emaciated face accentuated by lines in shades of brown and cream-white against a blue background and is cut by a more fluid band that seems to reverberate within the face. These three versions show the gradual transformation to an androgynous face.

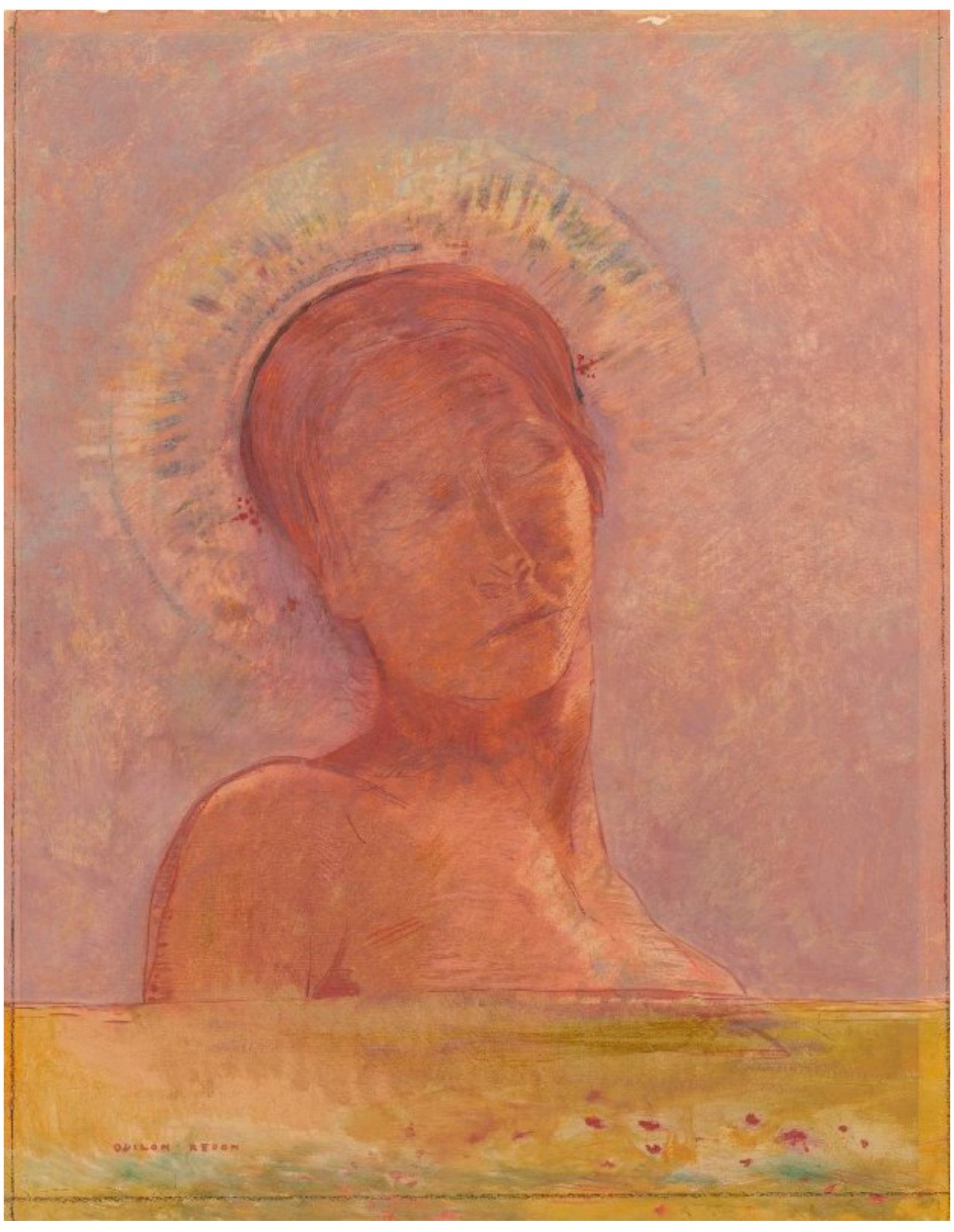

**Figure 3.** Odilon Redon, *In Heaven* or *Closed Eyes*, 1889. Thinned oil (*peinture à l'essence*) on orange wove paper on cardboard, 45 × 35 cm. Van Gogh Museum, Amsterdam (State of the Netherlands), s0500N1999.

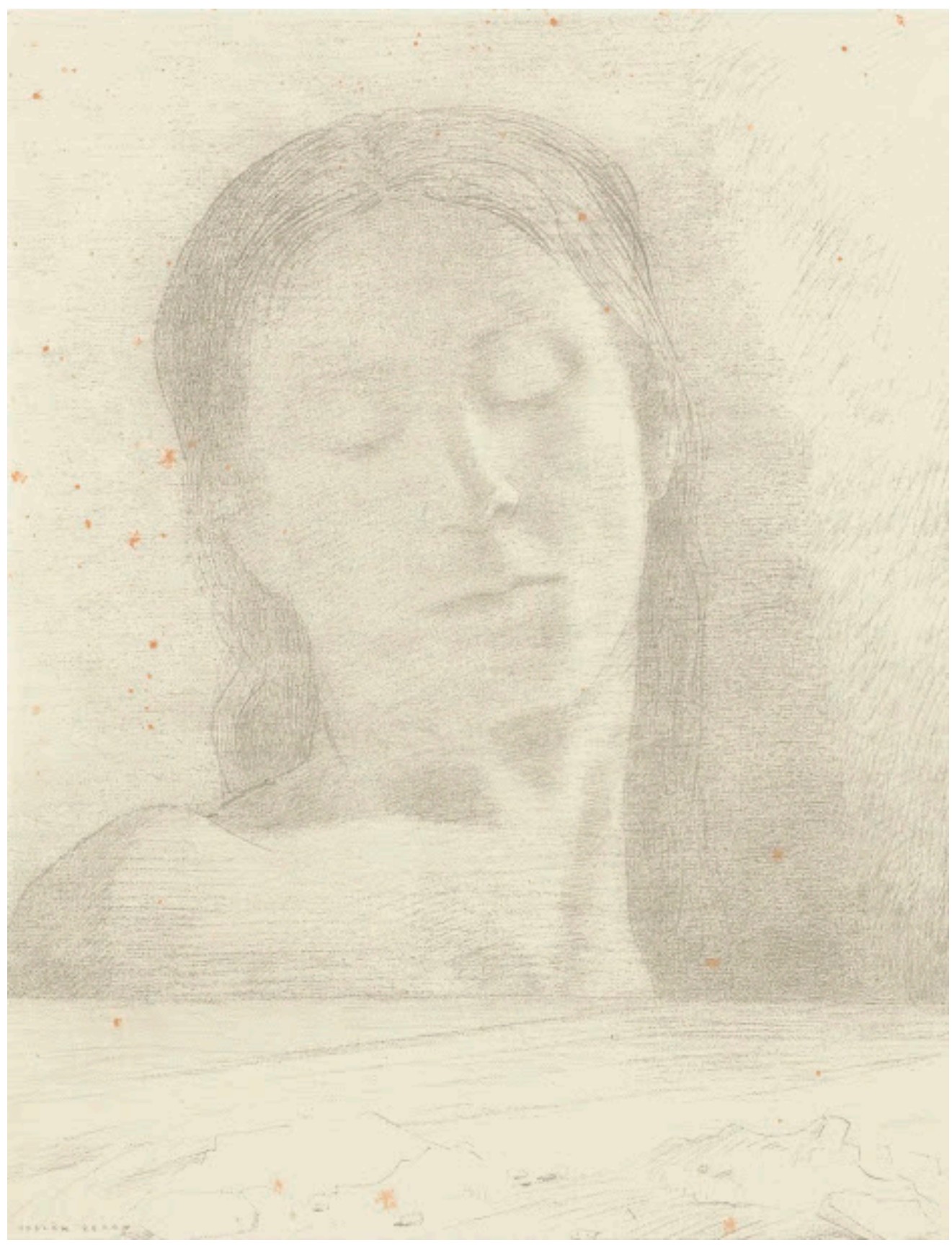

**Figure 4.** Odilon Redon, *Closed Eyes* (*Yeux clos*), 1890. Lithograph on chine collé on wove paper, 56.5 × 40.5 cm. Van Gogh Museum, Amsterdam (State of the Netherlands), p0880N1996.

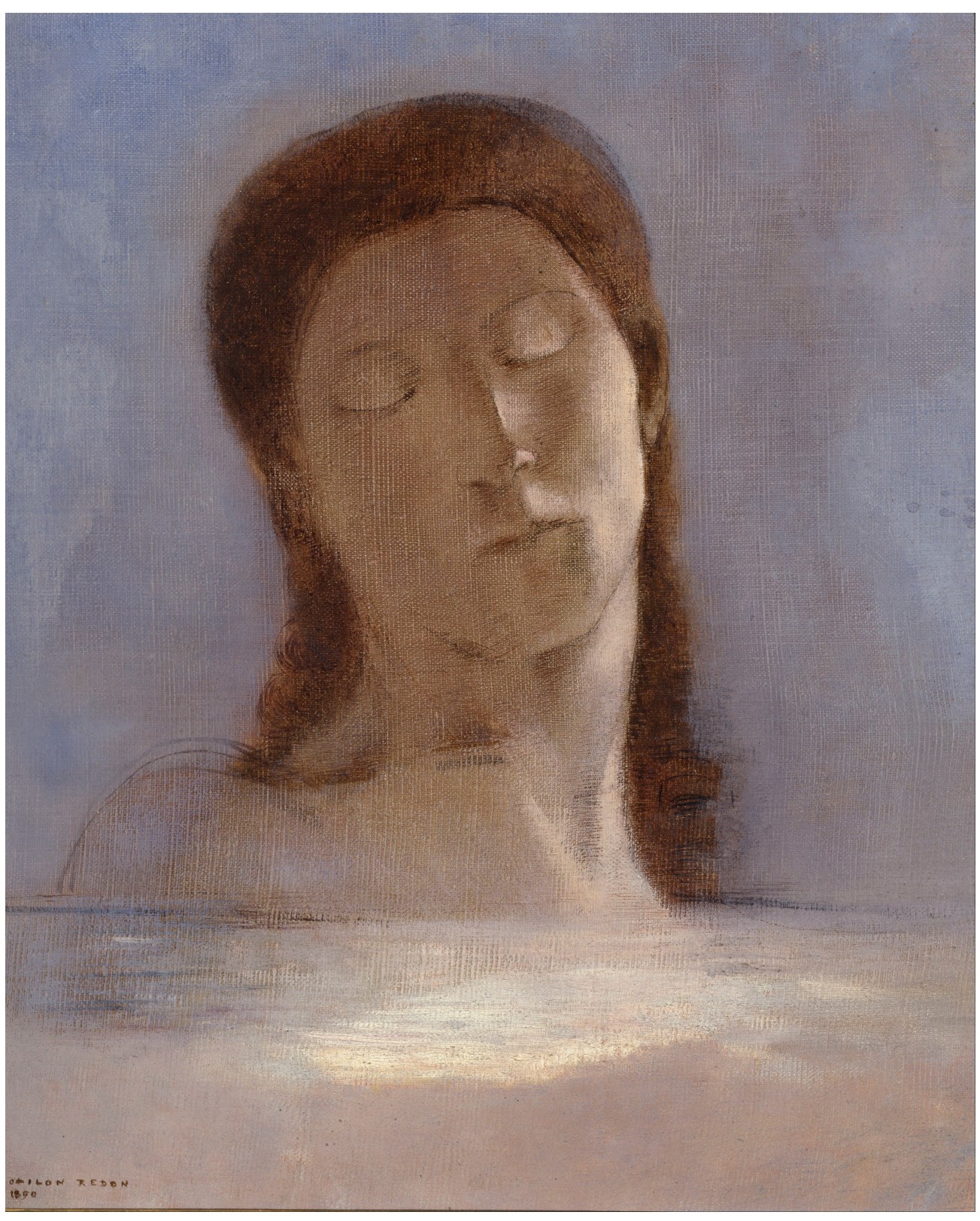

**Figure 5.** Odilon Redon, *Closed Eyes*, 1890. Oil on cardboard, 44 × 36 cm. Musée d'Orsay, Paris.

Critics have highlighted the ethereal nature of this isolated figure, supported by a floating realm of dreams and reverie. Belgian critic Edmond Picard dedicated an eponymous article to *Closed Eyes*, distinguishing it from the lithograph and describing it as "a very gentle head of a woman, leaning, with a calm face like death, yet with closed eyelids and

imperceptibly tightened lips, betraying a slumbering life, or rather, absorbed in a dream" (Picard 1890, p. 412). He likened this figure to a symbol of spiritual and abstract femininity, comparing it to a "work loved devoutly, an image of a saint, an image of a virgin, an image of femininity" (p. 412). The chaste and contemplative image leads the critic to speculate on the model's relation to the artist, seeing in it "the features of Madame Redon" (Rapetti and Redon 2011, pp. 228–29).

But the artist's response to this review is of interest, as Redon denied any connection between the figure and reality: "[It is an] overly laudatory article, surprising me with everything you saw in [the painting] and everything I unconsciously put into that androgynous head. You saw the features of Madame Redon! Perhaps. Making little use of living models, I sometimes reflect the faces around me; however, not for monsters" (p. 228). Yet the recent discovery of a preparatory sketch (de Carvalho 2022), inverted in format and preceding the painted versions, which likely served as the basis for the lithograph, indeed resembles Camille Redon, the artist's wife. The various stages of transformation of the work attest to a progressive process of synthesis: an initial feminine version in the first painting, followed by a more masculine version in the lithograph, and finally, an androgynous version in the final painting acquired in 1904 by the Musée du Luxembourg in Paris (now at Musée d'Orsay).

The mixed reception of this simple and mysterious series was captured at the time by the occultist Luc Hizarbin (likely a pseudonym), who attempted to disentangle Redon's creative process. Following the retrospective exhibition at the Durand-Ruel gallery in 1894, which featured the lithograph of *Closed Eyes*, the critic gave a striking explanation of the work in the esoteric journal *Le Voile d'Isis.* He argued that the artist traverses different states of consciousness, semi-consciousness, and unconsciousness in a "continuous, graded series" (Hizarbin 1894, p. 2), leading to other the figures sinking into the ether. The term "unconscious", coined by Edouard von Hartmann in his 1878 philosophical treatise, accorded significant importance to artistic creation. Hizarbin adopted this principle, distinguishing two levels of interpretation based on the gendered metaphor of creative inspiration. While the conscious state leads the artist's receptive body to be inspired by destiny, it can only sketch a form in development. Conversely, the unconscious state amplifies the visionary capacities of the artist who sees the beyond, placing their body as an active agent in creating. The androgynous model serves as an intermediary to this inverted schema: the artist transforms the passive and feminine consciousness of inspiration into the active and masculine unconsciousness of creation.

We do not know if Redon was aware of Hizarbin's text, but the artist did emphasize the role of the unconscious as a dual engine of inspiration and creation in his *Journal*: "Imagination is this sovereign that suddenly opens up magnificent, surprising seductions and captivates us. It has been my guardian angel. Formerly much more than today, alas... It is also the messenger of 'the unconscious,' that very high and mysterious character... Nothing in art is done by will alone. Everything is done through docile submission to the arrival of 'the unconscious'" (Redon 1923, pp. 34–35). This remark resonates with the symbolism—which suggests more than it represents—contained in these indeterminate figures.

A final comment by Picard attests to this dual understanding of the androgynous. The critic wrote that Redon's fifty lithographic copies were primarily intended "for you Aesthetes, for you alone" (Picard 1890, p. 402). The term "aesthete" directly refers to the aestheticism of English Pre-Raphaelite artists, who had been promoted in Belgium by the critic Émile Verhaeren. Following his trip to England in 1882, Verhaeren produced a series of reviews of the Royal Academy exhibition for the journal *L'Art moderne,* associating painting and literature to androgynous aesthetics: "Revolutions in painting have always had a literary side in England. The Pre-Raphaelites were inspired by the *Vita Nuova* just as the aesthetes [were inspired by Théophile Gautier's] *Mademoiselle de Maupin*. Fashionable books explain artistic trends" (Verhaeren 1885, p. 297). The reference to Gautier's work, which featured bisexuality and cross-dressing, was meant to show that the aesthetic passion

for androgyny was addressed to men who nurtured a sensitivity for double figures and gender ambiguity.

Verhaeren solidified his symbolist theory in 1887 in relation to the work of Fernand Khnopff. This painter's ambiguous aesthetics relied, according to Verhaeren, on the suggestion of "pagan symbolism" (Verhaeren 1989, p. 232) inherited from the poetry of Stéphane Mallarmé and materialized as enigmatic figures: "The Symbol thus always refines itself, through an evocation, into an idea: it is a sublimation of perceptions and sensations; it is not demonstrative but suggestive; it ruins all contingency, all facts, all details; it is the highest expression of art and the most spiritualist" (232). In homage, Khnopff offered Verhaeren a vision of this ideal in *With Verhaeren. An Angel* (1889) (Draguet 2018, pp. 185–90). The warrior's armor and the sphinx with a feline body, being hypnotically caressed to sleep, protect the androgynous body with a feminine face. The artist chose to evoke a Hellenic sphinx rather than the original Giza sphinx, further deepening the sexual enigma created through a gaze, both from the figures and the viewer, suspended by fascination and repulsion.

Khnopff primarily used his sister Marguerite and close family acquaintances as models, particularly the sisters Elsie, Lily, and Nancy Maquet. This choice followed the early collaboration between Khnopff and Péladan, who in 1884 asked the artist to illustrate his novel *Vice Suprême.* From a first drawing representing the heroine of the novel, Leonora d'Este, exhibited at the Salon des XX in 1885, a scandal ensued from which Khnopff would never recover (Draguet 2018, pp. 77–79). The opera singer Rose Caron, a Wagnerian muse of the time, recognized herself in the androgynous traits imagined by the artist and cried scandal. Péladan took up the cause for the artist, who decided to destroy the work, emphasizing the fleeting nature of Caron's androgynous existence on stage: "Madame Caron in *Fidelio* will understand the ideal existence of a third sex, not forgetting that its existence is the briefest, that it is less a species than a transitory state" (Péladan 2010, p. 67). This episode reveals the ambivalent union of the masculine and the feminine that unites art and life in a discontinuous aestheticism. From the living model to free inspiration, the symbolists constructed their figures from a purified language drawn from abstract symbols, which critics saw as a source of inspiration stemming from the ideal androgynous model.

## 4. From Self-Image to Public Visibility

Other symbolist portraits cast doubt on the sexual identity of their painters, giving the androgynous figure the power to transcend social taboos related to women's status and the forbidden passions associated with homosexuality. In a male-dominated context, few women artists were able to make a name for themselves (Foucher 2015). Jeanne Jacquemin, also known as Jeanne Coffineau, was in the symbolist cohort; she may have used her emaciated appearance for the figure of Christ or Saint George (Jumeau-Lafond 2003). A comparison with one of the few photographs of her confirms an air of introspection consistent with that of the spiritual martyr or warrior (Figure 6). Few works of this artist are known, usually from mentions in catalogs and critical reviews of the time. The work titled *Séraphitus-Séraphita* (unknown location), presented at the Brussels Salon des XX in 1893, recalled the androgynous theme of Balzac's novel *Séraphita* (Delevoy 1981, p. 192).

According to the critic Rémy de Gourmont, Jacquemin's pastel compositions were Baudelaire-inspired expressions of androgynous ambiguity: "a charming immorality that cares very little about specifying gender and leaves the doubt of androgyny floating like a haze of unhealthy and adorable desires around heads infinitely weary of living, expressed in pastels of a technical skill very rare in a woman" (de Gourmont 2006, p. 252). While her evocation of symbolism associated the artist with Rosicrucian aesthetics, she was not allowed to exhibit at the Salon de la Rose † Croix. Péladan prohibited women from presenting their work at the salon because "according to magical law, no work by a woman will ever be exhibited or executed by the Order" (Salon de la Rose † Croix 1894, p. 28). In a paradoxical display of sexism, Péladan denied women the opportunity to exhibit, although he was surrounded by progressive women as well as men. Jacquemin did participate in the second Exhibition of Impressionist and Symbolist Painters, held between May and August

1892 at the Le Barc de Boutteville gallery in Paris. The catalog (with an introduction by Aurier) did not mention her presence, but Gourmont mentioned her in his review: "Exiled from the Rose † Croix, where women were not admitted (although it abounds with more virile works), Madame Jacquemin has taken refuge with Mr Le Barc de Boutteville, where she exhibits some pastels" (de Gourmont 2006, p. 252).

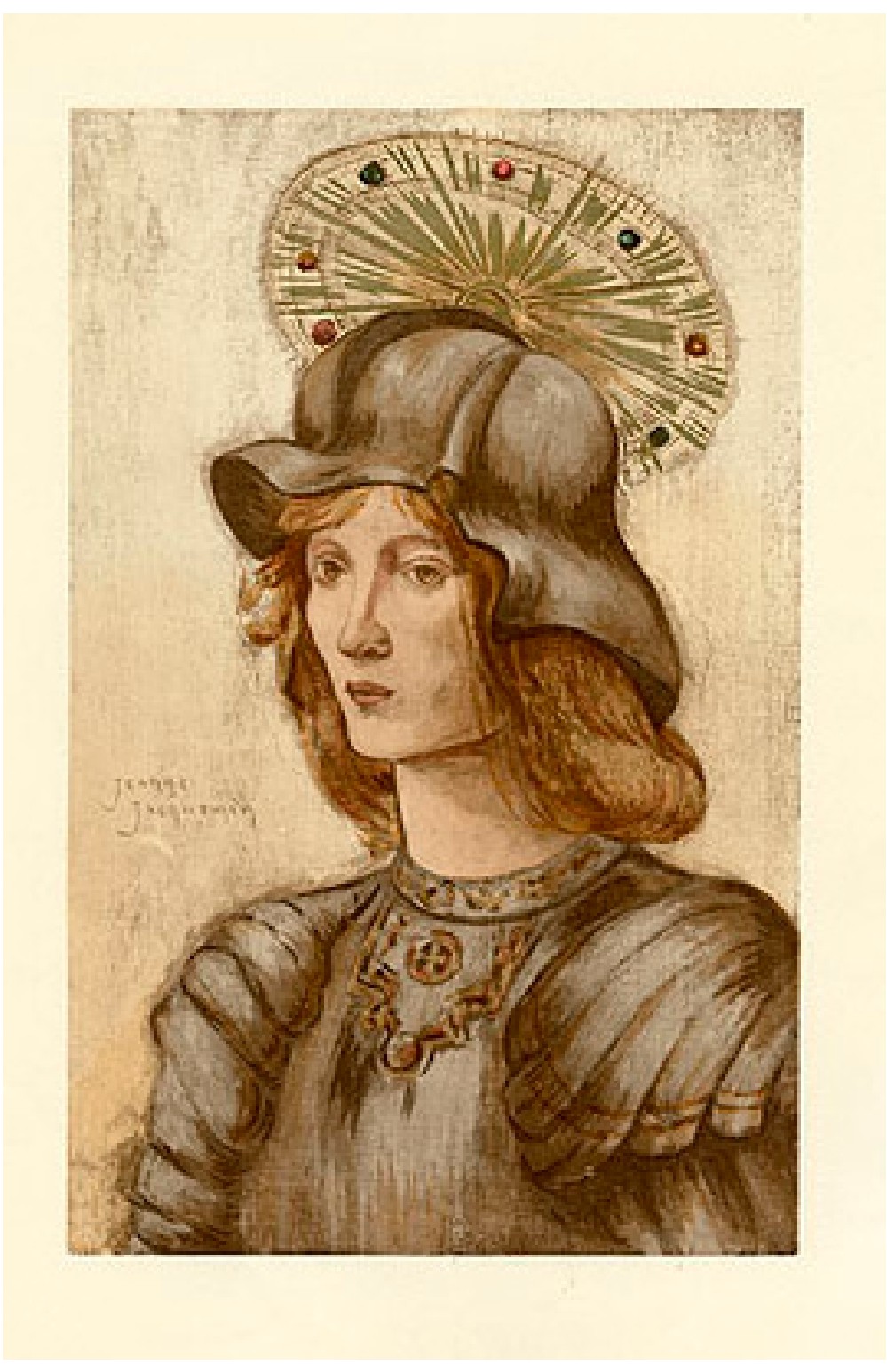

**Figure 6.** Jeanne Jacquemin, *Saint Georges*, 1898. Lithographic on wove paper, 40 × 30.5 cm, print from *L'Estampe moderne*, vol. I, Paris, F. Champenois, 1898, non-paginated. Creative Commons attributed.

Jacquemin's symbolism is a rare example from a female artist within these male circles, making the androgyne a true figure of personal emancipation. The symbolist poet Saint-Pol-Roux dedicated a poem to her titled "Le sexe des âmes" ("The Sex of Souls"), in which he explored the creative act. He described the inversion of sexes occurring in the darkness, where the soul of the woman becomes a faun and that of the man, a nymph. The narrator of the poem ponders: "Is the soul of the woman of the female sex? Is the soul of the man of the male sex?—Is the sex of the soul of the man and that of the woman masculine or feminine?" (Saint-Pol-Roux 1901, p. 267). The ambivalence of symbolist and idealist circles lay in these idealistic explorations of gender inversion and fusion, associated with artistic strategies in which artists adopted androgynous attire.

Contesting figurative norms, on the dual plane of resemblance and gender identification, ultimately intertwined with the early homosexual sociality of the late 19th century. The feminine androgynous ideal embodied by Jacquemin found a counterpart in the young Dutch artist Léonard Sarluis, who came to train in Paris in 1894 (de Pawlowski). During a literary banquet in honor of Verhaeren on 25 February 1896, French symbolists associated young Sarluis with the embodiment of their ideal. The writer and art critic Jean Lorrain presented him descriptively: "This Batavian artist, a living Giotto if rumors are to be believed, has been revolutionizing the aesthete camp for a month... the smile of a da Vinci, the eyes of a Donna Ligeia, the neck of a Gabriel-Dante Rossetti, the talent of Michelangelo, and a virgin on top of that... strolls through life in a black velvet doublet, in long gloves of white suede embossed with gold embroidery, with the curly and too heavy hair of Florentine pages on his shoulders..." (Lorrain 1896, p. 2). The young artist captured all the attention because "like an ancient Eros, he enchantingly unsettles men and women" (p. 2).

In the defense of an ideal aesthetic, Sarluis became the model for the poster of the fifth Salon de la Rose ✝ Croix, held between March and April 1896, painted by Armand Point. The young artist was dressed as a helmeted figure in Perseus's armor, holding the heads of Émile Zola and the Gorgon Medusa. On a crusade against realist aesthetics, Péladan aimed to revitalize his salon, which had been much criticized, notably by Zola, who denounced its "lamentable overflow of mysticism" (Zola 1896, p. 1). Zola lambasted the physical and moral degeneration of androgynous aesthetics: "These sexless virgins with neither breasts nor hips, these girls who are almost boys, these boys who are almost girls, these larvae of creatures emerging from limbo, flying through pale spaces, agitating in confusing regions of gray dawns and twilight the color of soot!" (p. 1).

The poster showed the close connection between the artist and his model. A "student of Armand Point" (Salon de la Rose ✝ Croix 1896, p. 18), Sarluis undoubtedly inspired Point in his composition entitled *L'Androgyne* (p. 18), a fresco portrait numbered 85 in the salon catalog. Only an undated sketch in red chalk of this painting remains (Figure 7). It depicts a young boy with evanescent hair parted in the middle, and treated in the sfumato style of Leonardo da Vinci; it resembles a later photographic portrait of Sarluis. At the same Salon, Sarluis exhibited *Jeune Florentin* (p. 19), portrait number 93 (location unknown), whose egg tempera technique was inspired by Point and likely has not survived.

In line with Zola's severe critique, Lorrain considered Rosicrucian aesthetics in relation to psychopathology. He claimed that the series of Renaissance art pastiches on display concealed other intentions, especially in (referring to Sarlius) "the work of this young painter Léonard, like da Vinci, Sâr, like Péladan, and personal like a mirror where the most dangerous qualities of Tiepolo, Veronese, and Peter Paul Rubens would be reflected in turn" (Lorrain 2007, p. 298). Lorrain amalgamated the androgynous ideal in these portraits with concepts of sexual deviance spread by eugenics theories of the time: "I won't say anything about the shameful portraits exhibited by this disconcerting young male painter; males and females fall within the clinic of Dr. Lombroso" (p. 299). Using irony, Lorrain announces the end of the symbolist project alongside the aestheticism of the English Pre-Raphaelites through the metaphor of a stock market crash: "The crash of [Edward] Burne-Jones... Burne-Jones has in his camp the aesthetes of Avenue Henri-Martin and the purple socks (because the blue stockings in this neighborhood wear purple socks) of the Plaine Monceau"

(p. 285). By associating feminist currents (blue stockings) with aesthetes, Lorrain equated the symbolist ideal of the androgyne with societal changes he believed were leading to the downfall of art: "the crash of Burne-Jones, the crash of Botticelli, yes, of Botticelli himself, now outdated, become suspect and compromised thanks to Maurice Denis, Mrs. Jacquemin, and Mr. Armand Point" (p. 188).

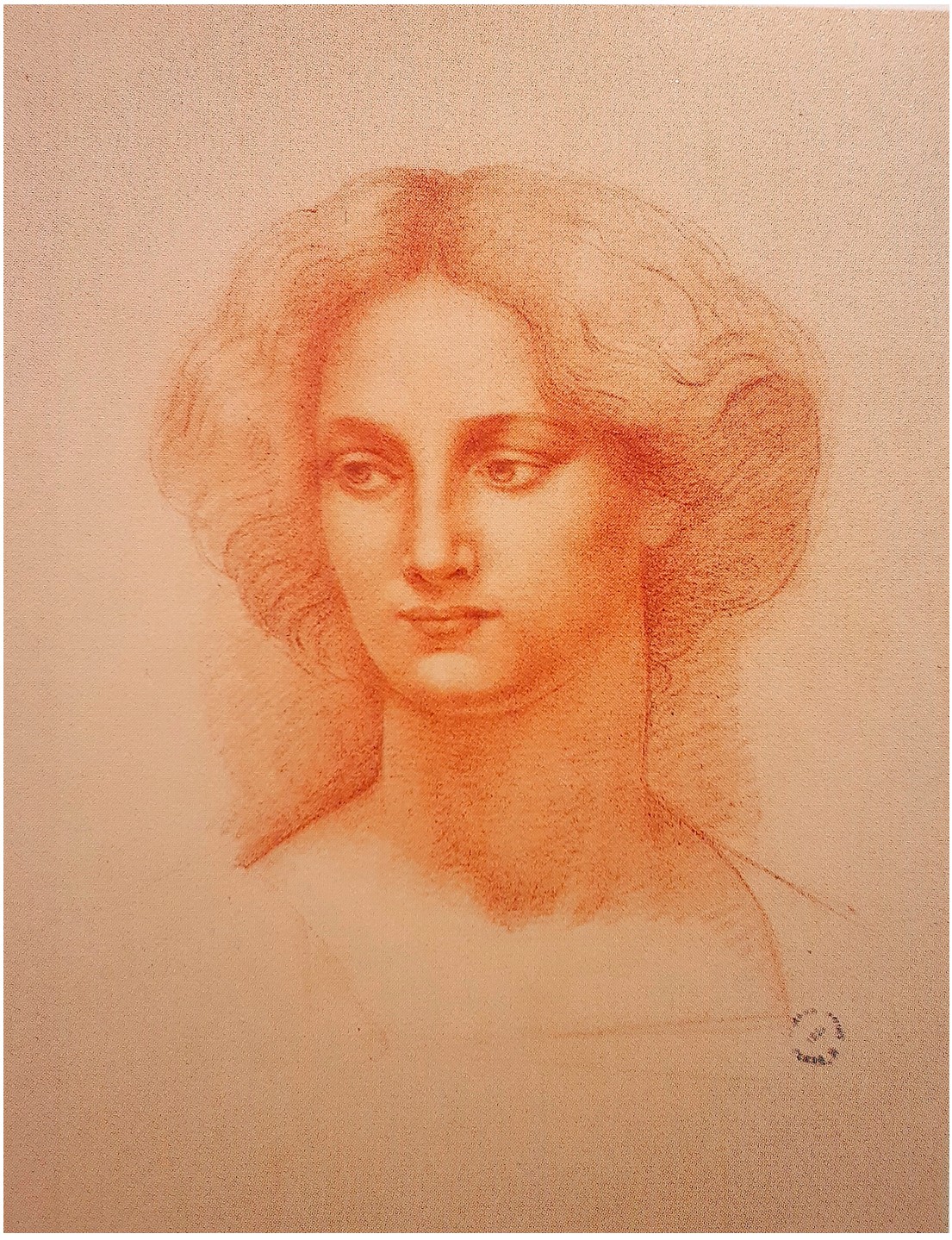

**Figure 7.** Armand Point. *Androgyne.* 1899. Red chalk on paper. 43 × 34 cm. Private collection, Lucile Audouy.

### 5. From Androgyny in Freud and Beyond

Symbolist and idealist androgynous aesthetics could have remained a historical curiosity if they did not mark the origins of a proto-queer vision in art history. Following the androgynous representations of symbolist painters, the role of sexuality in the creative process, fiercely debated at the time, led to Sigmund Freud's famous interpretation in 1910 of Leonardo da Vinci's childhood memory (Freud 1991). The Renaissance master resurfaced in the psychological interpretation of repressed homosexuality that Freud identified with the concealed form of the vulture in *The Virgin and Child with Saint Anne* (1503–1519). The psychoanalyst proposed a formal extrapolation of the hidden form in the folds of the Virgin's tunic, which he then associated with the memory of the bird that would have placed its tail in the artist's mouth. As highlighted by Meyer Schapiro, it is not the historical dimension (erroneous, notably due to mistranslation) but the heuristic character of the method that should interest us. Subsequent readings have emphasized the crucial contribution of this interpretation to the understanding of art in relation to artists' neuroses, the mechanisms of visual language in the collective unconscious, and as a tool for deconstructing gender norms, albeit one associated with a patriarchal vision but nevertheless embraced by feminist artists and theorists (Silverman 1992).

These aspects constitute a central moment in the construction of Freud's theory based on the androgynous imaginary derived from symbolist explorations and linked to a less deterministic conception of homosexuality. Aside from the theories of sexual inversion and the third sex discussed in German, French, and Italian journals of the time, Freud was particularly interested in historical models of the androgyne in connection with sexual psychology. He discovered a study published in 1903 by the Dutch physicist and theologian Lucien von Römer on the German sexologist Magnus Hirschfeld (von Römer 1903). von Römer described numerous representations of androgynous, hermaphroditic, and bisexual figures from Egyptian, Indian, Japanese, and Greco-Roman cultures, which very likely inspired symbolist artists during the same period. As Whitney Davis points out, it is possible to find the visual model of the vulture, improperly associated with a kite, in Freud's writings citing Römer, especially his study of the three-headed Egyptian goddess Mut, a fertility symbol carrying a phallus (pp. 735–736). Associated with the esoteric goddess Isis, the androgynous Mut is often depicted young, with feminine breasts and a male sex, which corresponds, according to Römer, to a balance between the active and passive power of fertility (p. 921). Römer argued that homosexual artists are often devoted to the androgynous symbol precisely because only they "are the most harmonious individuals" (p. 736).

Freud was struck by this detailed historical analysis, as it allowed him to model the enigma of art on that of bisexual androgyny. Freud postulated a common bisexual disposition of humanity across all cultures and historical periods, manifested in the androgynous imagery collected by Römer. In Whitney Davis's words: "the problem of sexual inversion was replaced by the problem of sexuality itself" (Davis 2010, p. 205). Building on studies of homosexuality developed in Magnus Hirschfeld's research circle (Taylor et al. 2017; Delille 2019), Freud nonetheless distanced himself from the conception of innate homosexuality (the theory of the third sex) and acquired homosexuality (moral perversion inherited from psychopathology). Androgyny did not correspond to the model of the homosexual body shaped by medical science in the 20th century, but rather to a nonbinary gender chimera that constantly oscillated between the sexes—even transcending them in chastity.

This heuristic dimension of the androgyne resonates with fin-de-siècle symbolist practices, as well as other publications linked to Hirschfeld's circle, which offered a vision of so-called "intermediate sexuality" in Europe, outside heterosexual norms. In 1899, the journalist Otto de Joux, in an article published in Hirschfeld's journal, briefly mentioned a group of French artists, whose mystical rites resembled homosexuality as described in Germany at the time: these artists were the Order of the Rose † Croix of "Marquis de La Rochefoucauld and Sar Joséphin Péladan" (de Joux 1899, p. 126). The journalist questioned

the meaning of the ambiguous first name Joséphin—between Joseph and Joséphine—which is associated with the rules of virtue and chastity imposed in this artistic circle: "The members must only be men who are completely pure, from youth to old age, who have never kissed feminine lips" (p. 126). The asceticism of the symbolists is thereby equated to a suspension of desire and a rejection of sexuality with women. This restrictive vision contrasts with Freud's theories of art history and the idiosyncratic dimensions of human existence, according to which gender identities can proliferate in artistic consciousness and unconsciousness and which, curiously, was never followed up on, even after the exploration of the unconscious in surrealism. Should the androgynous creations of the symbolists be considered as an alternative understanding to those proposed by Freud and the Hirschfeld circle in Germany? That is, as a proto-queer vision of art, based not solely on (homo)sexuality, but on a more fluid representation of gender, which would give rise to the disappearance of the masculine and feminine in the abstract art of the early 20th century?

## 6. Conclusions

Throughout the 20th century, numerous psychologists and art historians theorized on the "bisexual" nature of the artist. The passive experiences of imagination and inspiration are activated in the act of creation, the origin of the symbolic construction of the image. This conception is latent in the symbolist aesthetics of Aurier, Péladan, and Verhaeren, inspiring the androgynous figures imagined by symbolist artists such as Séon, Redon, Jacquemin, and Sarluis. The artist's relation to the model has been a driving force in the modern psychology of the artist, whose ambivalent symbols suggest a proto-queer vision related to the emergence of sexual theories. Writing a queer history of art is thus about reclaiming buried narratives and knowledges that have often been pathologized by the medical sciences. It is also about reinvesting in these queer genealogies (Lorenz 2012) that reveal the shadows, paradoxes, and tensions at the heart of a floating, uncertain, and shifting vision of queer artistic practices.

**Funding:** This work was completed with the support of LARHRA (UMR 5190), Lyon, France.

**Data Availability Statement:** The original contributions presented in the study are included in the article, further inquiries can be directed to the corresponding author.

**Conflicts of Interest:** The funding sponsors had no role in the design of the study; in the collection, analyses, or interpretation of data; in the writing of the manuscript, or in the decision to publish the results.

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
