# Peer review of "Symbolist Androgyny: On the Origins of a Proto-Queer Vision"

_arts, 2023_

Round 1

Reviewer 1 Report

Comments and Suggestions for Authors

A very interesting article. Carefully prepared and well-written. The selection of materials and bibliography is unobjectionable. I highly commend this text. Some sections could be written more synthetically.

Author Response

thank you for your expertise, which did not require any revision of the text. I enclose the file with minor revisions requested by other reviewers. The English language check is currently being processed (as I am not a native English speaker myself).

Reviewer 2 Report

Comments and Suggestions for Authors

This is a compelling paper about androgyny in Symbolist art and aesthetics. The reading of Odilon Redon's revisions of a portrait is particularly compelling. However, the evidence is sometimes stretched a little thin and the scope stretched a bit wide. 

On evidence, there is a mismatch at times between primary sources and interpretation. On Picard's interpretation of Redon, the authors state Redon's figure is "deemed a symbol of spiritual and abstract genderlessness" but the Picard quote specifically focuses on "a very gentle head of a woman" and "an image of femininity." This is hardly "abstract genderlessness" but rather a femininity. Perhaps we could say it is an asexual femininity, but that is different from saying it is genderless. 

This relates to the problem of scope. The article notes from the outset that it is hard to project "queer" backward in time because of the historicity of gender and sexuality. But this suggests that we may need something narrower than "queer." Most often, androgyny appears in the analysis as something valued because it transcends the sexual, which means it is less prototypical of queer in the sense of "homosexual" and more of queer in the sense of "asexual" or "gender nonbinary." As the article notes, through Whitney Davis, there was a subsumption of sexual inversion by sexuality. But that means that reading things as "homosexual" overemphasizes a contemporary sense of queer collapsed into "gay" instead of some other subset identity. Further, to what extent are androgyny and chastity separable, and if they are, when is the latter the better term for what is being studied? 

A further complication arises in the article's turn at the end to thinking about how androgyny is appropriated by Freud and Hirschfeld. The inspirations for Freud, as outlined in the essay, are da Vinci and a study by Lucien von Römer. But that makes it seem like the discussion of Symbolist artists that preoccupies the article is irrelevant. Similarly, Hirschfeld seems to make a quick reference to Joséphin Péladan in a journal article, but this is hardly evidence of resonance or influence. Does the article mean to say that the art historical discussion preceding provides a different mode of analysis than Freud or Hirschfeld? Is there an alternative method here to Freud's treatment of da Vinci? Again, it might be the refusal to reduce queer to "homosexual" that might be at stake. 

Finally, because the article's archive is Symbolist, the framing of "fin-de-siècle" is also too expansive, and the specific Symbolist context is better suited for the title of the essay and its introduction. 

Author Response

thank you very much for your detailed and very pertinent expertise. i have added and taken into account your remarks: 
- I've modified Picard's assessment of Redon's work;
- I have added and modified the last part devoted to the study of Freud and Hirschfeld, based on your wise suggestions. 
- I've changed the title

- The English language check is currently being processed (as I am not a native English speaker myself).

Reviewer 3 Report

Comments and Suggestions for Authors

I would suggest that the works quoted be just listed without further reference to their presence in the main text.

Author Response

thank you for your expertise. I don't understand how you want the works to be mentioned in the text: which reference should be removed (Date? Full name?). 

The English language check is currently being processed (as I am not a native English speaker myself). Minor revisions have been made in response to other reviewer assessments. 
